# Effect of Ceramic Capillary Parameters on Bonded Morphology and Strength

**DOI:** 10.3390/mi12010024

**Published:** 2020-12-29

**Authors:** Jun Cao, Junchao Zhang, Kexing Song, Baoan Wu, Yong Ding, Dingbiao Chen, Yutian Ding

**Affiliations:** 1School of Mechanical and Power Engineering, Henan Polytechnic University, Jiaozuo 454000, China; zjc_xlyx@sina.com; 2School of Materials Science and Engineering, Henan University of Science and Technology, Luoyang 471000, China; kxsong@haust.edu.cn; 3Chongqing Materials Research Institute Co., Ltd., Chongqing 400700, China; caolinc@163.com; 4Zhejiang Tony Electronic Co., Ltd., Huzhou 313008, China; diego.ding@tony-tech.com; 5Changzhou Hengfeng Special Conductor Co., Ltd., Changzhou 213000, China; pur@czhftc.com; 6School of Materials Science and Engineering, Lanzhou University of Technology, Lanzhou 730050, China; ytding@lut.cn

**Keywords:** ceramic capillary, geometric parameters, bonded point morphology

## Abstract

The effects of the geometry parameters of a ceramic cleaver on the morphology of ball and second bonded points were studied using an automatic wire bonder, push pull tester, scanning electron microscope, ceramic capillary with different geometric parameters and φ25.4 μmAg-5Au bonding alloy wire, etc. The result shows that when the inner hole diameter (IHD) of the ceramic capillary is 1.3 times the diameter of the alloy wire (33 μm), the neck morphology of the ball bonded point (first bonded point) meet the requirements. The neck of the ball bonded point appeared to fracture when the IHD is 26 μm; The neck of the ball bonded point appeared as an irregular shape when the IHD is 41 μm. When the inner cutting angle diameter (ICAD) is 64 μm, the size of the mashed ball diameter (MBD) is qualified. When the ICAD is 51 μm, the MBD is too large and mashed ball overflows the pad. When the ICAD is 76 μm, the ball bonded point is too high. When the inner cutting bevel angle (ICBA) is 100°, the MBD size meets the requirements of the pad. When the ICBA was reduced to 70°, the ball bonded point is eccentric. When the ICBA was increased to 120°, the MBD is too large and is connected to the adjacent pad contact. The size of the fish tail of the second bonded point (second bonded point) changed in the same direction as the tip diameter (TD) changes. When the TD is 178 μm, the fish tail shape is regular and symmetrical. When the working face angle (WFA) is 8° and the outer circular radius (OCR) is equal to the diameter of the alloy wire (25.4 μm), the fish tail shape is regular. When the WFA is higher than 11° or the OCR is higher than 30 μm, the fish tail will appear as virtual welding, and when the WFA is less than 4°, the fish tail of the second bonded point will break due to thinning. When the OCR is less than 20 μm, the fish tail of the second bonded point is too long and causes a short circuit.

## 1. Introduction

Bonding is one of the key technologies in microelectronics [1]. As the key tool in wire bonding, the geometric parameters of the ceramic capillary determines the quality and reliability of the chip packaging. Therefore, the choice of ceramic capillary in the field of microelectronics is particularly important for wire bonding [2]. With the booming development of the microelectronics industry, the demand for ceramic capillaries is also increasing [3,4,5]. At the same time, the application technology of the ceramic capillary in wire bonding needs to be further improved [6,7,8,9,10]. Many scholars have conducted in depth explorations on the composition, structure, motor history [11], vibration characteristics [12] and other aspects of the ceramic capillary, which provide a valuable reference for the parameter design of the ceramic capillary, and promote the more efficient application of the ceramic capillary in the field of microelectronics. For example, Yao Li et al. [13] found that if the outer circular radius (OCR) is too large, the length of the bonded point will be too small, hence the bonded points will be weak. On the contrary, if the OCR is too small, the length of the bonded points will be too large, resulting in the interconnection of adjacent bonded points and the package failure. Goh K S et al. [14] found that the cone core angle, the inner cutting bevel angle (ICBA) and the inner cutting angle diameter (ICAD) have a significant effect on the formation of deformed golden balls. A ceramic capillary with a small cone core angle and large ICBA can reduce the diameter of the gold ball by 12%. Kim I J et al. [15] studied the effect of modified doping on the ceramic capillary. The experimental results show that the change of micro grain size improves the strength, hardness, Young’s modulus, toughness and surface strength of ceramic capillary. Cao Jun et al. [16] studied the influence of bonding parameters on the bonding performance of copper wires, concluded that excessive ultrasonic power and bonding pressure caused cracks in the bonded points, and insufficient ultrasonic power would result in false welding. Qin Wen et al. [17] and Xue Rui et al. [18] studied the effect of bonding process parameters on the bonding quality of gold wire by using the finite element method and experimental method respectively, and improved the production process. The above-mentioned research mostly focused on the composition, structure, bonding parameters and other aspects of the capillary, while the influence of geometric parameters of the ceramic capillary on the bonded point morphology and bonding quality is rarely discussed. In this paper, by contrasting and researching the different geometric parameters of the ceramic capillary, the bonding experiment is carried out with φ25.4 μm Ag-5Au bonding alloy wire, and the influence of the geometric parameters of the ceramic capillary on the morphology of the alloy wire ball bonded point and second bonded point is analyzed. Additionally, we provide a theoretical basis for the optimization of the geometric parameters of the ceramic capillary.

## 2. Materials and Methods

### 2.1. Test Materials

The bonding wire material is φ25.4 μm Ag-5Au bonding alloy wire, and the composition and mechanical properties of alloy wire are shown in Table 1. The different geometric parameters of the ceramic capillary are shown in Table 2. 

The structure of ceramic capillary is very precise and complex [19,20], as shown in Figure 1.

### 2.2. Test Method

The bonding test was performed on the KAIJO FB-988 automatic bonding machine using φ25.4 μm Ag-5Au bonding alloy wire, N_2_ gas protection is used in the bonding process, and the gas flow rate is 0.6 L/min [22]. Wire bonding can be divided into ball bonding and second bonding according to the bonded point shape [23,24,25,26,27]. The bonding parameters of the ball bonded point are ultrasonic power 60 mW, bonding pressure 45 g; The bonding parameters of the second bonded point are ultrasonic power 95 mW, bonding pressure 75 g. Using the method of controlling variables, under the condition of the ultrasonic power, bonding pressure and other process parameters remain unchanged, we changed the IHD, ICAD, ICBA, tip diameter (TD), working face angle (WFA), outer circular radius (OCR) of the ceramic capillary, and then conducted a grouping test. A JEOL JSM-6700F scanning electron microscope was used to research and analyze the morphology of the ball bonded points and second bonded points. A Dage Series 4000, BS250 tester was used to test the ball shear strength and pull strength of the ball bonded points, and to analyze the influence of different geometric parameters of the capillary on the ball shear strength and pull strength. The ball shear strength test is shown in Figure 2, and the tensile test is shown in Figure 3. The tensile test fracture positions are A, B, C, D, where the break at point A is the ball detachment, the point B is the broken neck, the point C is the broken middle, and the point D is the neck fracture of the second bonded point. Breakpoints at point A, B, and D are all abnormal breaks, and breakpoints at the middle point of C and D are normal breaks.

## 3. Results

### 3.1. The Research on the Influence of IHD on the Morphology of Bonded Point

A free air ball (FAB) is formed by bonding alloy wire under the action of ball burning current [28], and the diameter of the FAB is generally 2–3 times of wire diameter [29,30,31,32,33]. The diameter of the FAB is 66.76 μm (2.63 times the diameter of alloy wire) in the test, which meets the test requirements. As shown in Figure 4, the FAB changes into the shape of the ball bonded point formed under the action of bonding power, pressure and temperature by force of a ceramic capillary. Figure 5 shows the morphology of the ball bonded points with the different IHD. When the IHD is 33 μm, the shape of the neck of the ball bonded point is regular and symmetrical, and the perpendicularity with the bonded point is better; the ball bonded point has a thin neck and irregular shape when the IHD is 26 μm. The neck of the ball bonded point deviates greatly from the center when the IHD is 41 μm.

The IHD of the ceramic capillary is a key parameter for the formation of the neck of the ball bonded point, which determines the morphological characteristics and strength of the neck of the ball bonded point. During the bonding process, the bonding alloy wire forms a neck feature perpendicular to the ball joint under the action of the capillary. The neck with high perpendicularity has less internal stress, the neck strength of the ball joint is high, hence the probability of neck breakage after bonding is low.

Figure 6 shows the distribution of the pull strength of the ball bonded points when the IHD of ceramic capillary is 26, 33 and 41 μm, respectively. The minimum values are 2.68, 11.26 and 8.11 g, respectively. Only when the IHD is 33 μm the ball bonded point tension has the largest average value—11.26 g, and the fluctuation range of the tension value is smallest. In the other two cases, the average value of pull strength is small and the distribution range is large, so the bonding strength is not good enough.

When the IHD of the capillary is 26 μm, the gap between the inner hole wall of the capillary and the alloy wire is 0.5 μm. The friction force increases between alloy wire and inner hole wall during the bonding process, which causes the ball bonding neck to shift, and the deformation rate increases, and then the pull strength value of the ball bonded point decreases. The neck of the ball bonded point is fractured due to the large deformation rate when the friction force is further increased, as shown in Figure 5d, even the partial tension value is heavily reduced, as shown in Figure 6. When the IHD of the ceramic capillary is 41 μm, the gap between the inner hole wall of the ceramic capillary and the alloy wire increases due to the large IHD, resulting in a low coaxiality between the alloy wire and the inner hole of the ceramic capillary and the neck of the ball bonded point is not symmetrical due to the alloy wire is not fixed enough during bonding. Finally, it leads to a decrease in the strength of the neck of the ball bonded point during service. When the IHD of the ceramic capillary is 1.3 times (33 μm) of the wire diameter, the neck shape of the bonding ball bond is regular and the symmetry is good, which meets the requirements of ball bonding characteristics.

Figure 7 is a statistical diagram of the position of the bonded point breakpoints after the bonding test of the ceramic capillary with different IHDs. It shows that only when the IHD is 33 μm the bonding strength is highest and almost all breakpoints (99%) are in the middle of the bonding wire. The breakpoints at point B indicates that the bonding strength between the ball bonded point and the substrate is lower than that of the bonding wire itself. The above data verify the conclusion that the bonding strength of the ball bonded point will decrease—whether the IHD of the ceramic capillary is too large or too small.

### 3.2. The Research on the Influence of the ICAD on the Morphology of the Bonded Point

Figure 8 is the morphology of the ball bonded points with. When the ICAD is 64 μm, the MBD size is distributed inside the pad thus meet the bonding requirements; when the ICAD is 51 μm, the MBD size is too large and overflows the pad, causing a short circuit and device failure; When the ICAD is 76 μm. the MBD bottom of the bonded point is too high, which reduces the radial size of the MBD and the bonding area between the bottom of the bonded point and the pad. The shearing force and reliability of the bonded point are reduced.

The bonding strength between the bonding wire and the pad is determined by the contact surface between the bonded point and the pad. Increasing the contact area between the bonded point and the pad is beneficial to improve the bonding strength of the bonding interface. The ICAD of the capillary determines the morphology of the ball bonded point. Within the size of the pad, the MBD of the ball bonded point, the contact area between the bonded point and the pad, and the shear strength of the bonding ball all increase with the decrease of the ICAD. As shown in Figure 9, when the ICAD is 51, 64 and 76 μm, the average shear strength is 49.67, 62.71 and 63.20 g, respectively. Figure 8d shows that the ball bonded point will be heavily deformed due to the too small ICAD (51 μm), and even overflow the pad to cause a short circuit and device failure. When the ICAD is larger than 76 μm, the height of the ball bonded point will increase more, and due to the volume of the ball bonded point remains unchanged, so the MBD of the ball bonded point decreases and the bonding strength decreases. In addition, the high MBD is easy to cause neck fracture and lead to bonding failure in the process of high density packaging. Only when the ICAD is 76 μm, the morphology and strength uniformity of the ball bonded point are the best.

### 3.3. Research on the Influence of ICBA on Bonded Point Morphology

The Figure 10 shows the morphology of the ball bonded points with the different ICBA. It can be seen from the Figure 10a that the MBD is distributed within the size of the pad and the bonded point morphology is good when the ICBA is 100°. When the ICBA is 70°, the MBD is within the size of the pad, but the contact area with the pad is small and the shear strength and the bonding strength is low. When the ICBA is 120°, the MBD size obviously exceeds the pad specification, causing the adjacent ball bonded points to contact, the bonded point to short circuit and the device invalidate.

Figure 11 shows the ball shear strength and pull strength of the ball bonded point when the ICBA of the ceramic capillary is 70°, 100° and 120°, respectively (all breakpoints are between Figure 10C,D in this test). It can be seen from Figure 11 that the average shear strength is 51.99, 63.50 and 70.56 g respectively, and the average pull strength is 6.87, 7.48 and 7.29 g, respectively. It can be seen that, with the increase of ICBA, the contact area between the ball bonded point and the pad increases, so the shear strength of the ball bonded point increases. However, the neck of the ball bonded point is not affected by ICBA. Therefore, the pull strength of the ball bonded point remains basically unchanged.

With the increase of the ICBA of the capillary, the MBD size, the contact area between the ball bonded point and the pad, the shear strength of the ball bonded point all increase, and the ball bonded point has good reliability. However, in the case of the ICBA being greater than 120°, the MBD of the ball bonded point will be too close to the adjacent ball bonded points, which results in a short circuit of the bonded point, as shown in Figure 10c. When the ICBA decreases, the height of the ball bonded point increases and the probability of false welding increases because the FAB volume remains unchanged and the MBD of the ball bonded point decreases. In addition, the height of the ball bonded point increases, the neck deformation rate of the ball joint becomes larger, and the stress concentration is more obvious, which reduces the reliability of the ball bonded point; When the ICBA is less than 70°, the control degree of the tip of the capillary on the FAB position is reduced, and the eccentric ball is easy to form in the bonding process, the bonding strength is reduced, and the short circuit probability of the ball bonded point is increased, as shown in Figure 10d.

### 3.4. Research on the Influence of TD on Bonded Point Morphology

Figure 12 shows the morphology of the second bonded points with the different TD. When the TD is 178 μm, the second bonded point is in the shape of a symmetrical fish tail and distributed with a small power ring mark; when the TD is 127 μm, the second bonded point is similar to a fish tail shape, but has less extension on both sides and no power ring mark; when the TD is 241 μm, the second bonded points are regular, the fish tail shape is symmetrically distributed, and the power ring mark is complete.

The tip of the capillary is the main stress surface that forms the second bonded point, and the TD of the capillary is one of the main factors that form the morphology and bond strength of the second bonded point [34]. In Figure 13, as the TD increases, the fish tail shape after bonding is larger, the contact area between the second bonded point and the pad is increased, and the bonding strength and reliability of the bonded point are improved. However, due to the limited distance between the pads, the TD is too large to easily contact the adjacent bonding alloy wires during the bonding process, causing mutual interference in the bonding process, reducing the bonding efficiency and bonding accuracy. Reducing the TD of the ceramic capillary is beneficial to the bonding of fine pitch pads, but when less than 127 μm, it will cause the second bonded point to be too small, and the contact area between the second bonded point and the pad will decrease, which easily causes bonded points to fall off and the bonded point to fail. Therefore, in the bonding process, within the allowable range of the pad size, a ceramic capillary with a larger TD should be selected as much as possible. For the φ25.4 μm Ag-5Au bonding alloy wire, when the TD is 241 μm, it has a good bonded point morphology: symmetrically distributed fish tail shape, completed power ring mark and better bonding strength.

### 3.5. Research on the Influence of WFA on Bonded Point Morphology

Figure 14 shows the second bonded points morphology with the different WFA of the ceramic capillary. The second bonded points are fish-tailed and symmetrically distributed, and produce semicircular power ring mark when WFA is 8°; when the WFA is 4°, the second bonded point is in the shape of an asymmetric fishtail, but its power ring mark is circular; when WFA is 11°, which the second bonded point has an asymmetric fishtail shape and there is no power ring mark.

The smaller the WFA of the ceramic capillary is, the greater the vertical bonding force pressure applied to the second bonded point on the working face, and the ultrasonic energy transmission also increases. The combined effect of the bonding alloy wire at high temperature, bonding pressure and ultrasonic energy mechanical coupling occurs between the bottom and the substrate metal, the length of the power loop increases [35], and the bonding strength of the second bonded point is improved. Figure 15 shows the pulling force value of the second bonded point of the bonding alloy wire with different WFAs, when the capillary WFA is 8°, the bond alloy wire second bonded point tensile value is the largest, with an average value of 6.82 g. When the capillary WFA is 4° and 11°, the average bonding alloy wire second bonded point pull strength is 4.88 and 4.71 g respectively.

When the WFA is less than 4°, it will reduce the force area of the fish tail of the second bonded point, resulting in greater force on the fish tail, making the fish tail too thin, that reduces the bonding strength and even breaks at the tip edge of the second bonded point failure, as shown in Figure 14d. Increasing the WFA, the fishtail thickness of the second bonded point and the bonding area will also increase, thereby improving the strength of the second bonded point and the reliability. However, when the WFA is greater than 11°, the vertical downward bonding area of the ceramic capillary to the second bonded point will increase more, while the pressure at the fish tail, the deformation rate and the bonding strength will decrease, causing virtual welding and even resulting in the bonding being invalidated.

### 3.6. Research on the Influence of OCR on Bonded Point Morphology

Figure 16 shows the morphology of the second bonded point with the different OCR of the capillary. When the OCR is 25.4 μm, in which the second bonded point has a fish-tail shape, the thickness gradually increases from the ceramic capillary working surface to the tip edge. When the OCR is 20 μm, the second bonded point is in the shape of a fish tail, but the shape variable is larger. When the OCR is 30 μm, the second bonded point has a fish-tail morphology, but the effective bond length between the fish tail and the pad is relatively short, so the bonding strength is low.

The OCR of the ceramic capillary determines the transition characteristics from the working face of the ceramic capillary to the tip edge. During the bonding process, the bonding alloy wire is deformed under the action of the outer circular, and gradually transitions from flattening and bonding to a complete bonding alloy wire [36]. The tension values of the second bonded points of the capillary when the OCR is 20, 25.4 and 30 μm, respectively, are shown in Figure 17. The average values of pull strength are 4.89, 6.56 and 5.59 g, respectively. The larger the OCR of the capillary, the smaller the area of the bonding alloy wire affected by the top of the capillary during the bonding process, the shorter the length of the second bonded point formed, and the longer the second bonded point length. When the OCR is 30 μm, the deformation area of the second bonded point will be larger, the effective contact area between the bonded point and the pad and the bonding strength will be smaller; When the OCR is 20 μm, the force area of alloy wire second welding point decreases. The fish tail area and shape variable of second welding point increases, but its effective bonding area with the substrate decreases, resulting in the reduction of bonding tensile value. Furthermore, it is easy to contact with adjacent bonded points, even causing short circuits and package failure due to the increase in fishtail shape and area, as shown in Figure 16d. When the OCR is 25.4 μm, that is, when the OCR is equal to the diameter of the bonding alloy wire, the bonding morphology is more regular and the bonding strength is higher.

## 4. Conclusions

In this paper, by studying the influence of the geometric parameters of the ceramic capillary on the morphology and strength of the 25.4 μm Ag-5Au bonding alloy wire ball bonded point and second bonded point, the following conclusions are drawn.(1)When the diameter of the inner hole of the ceramic capillary is 1.3 times (33 μm) of the diameter of the bonding alloy wire, the morphology of the MBD neck of the ball bonded point meets the requirements, and the increase or decrease of the IHD will cause the MBD neck of the ball bonded point irregular and reduced tensile strength.(2)When one of the ICAD or the ICBA of the ceramic capillary remains unchanged, the MBD size of the ball bonded point increases with the decrease of the ICAD or the increase of the ICBA. When the ICAD and ICBA is 64 μm and 100°, respectively. The size of the ball joint MBD meets the specifications.(3)The bonding strength and the contact area between the second bonded point and the pad increase with the increase of the TD of the capillary. However, the TD is too large to make the capillary contact the adjacent bonding wire, reducing the bonding accuracy. When the TD is 241 μm, the second bonded point has a good shape.(4)When one of the WFA and the OCR of the ceramic capillary remains unchanged, the fishtail shape of the second bonded point decreases with the increase of the WFA or the OCR. When the WFA and the OCR are respectively at 8° and 25.4 μm, the second bonded points are in a regular and symmetrical fishtail shape, and the power ring imprinting is complete.

## Figures and Tables

**Figure 1 micromachines-12-00024-f001:**
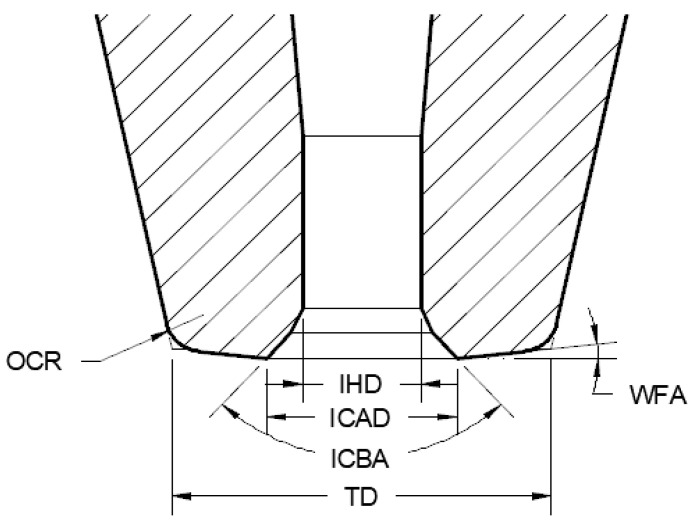
The main parameters of the ceramic capillary: IHD; ICAD; ICBA; TD; WFA; OCR [21].

**Figure 2 micromachines-12-00024-f002:**
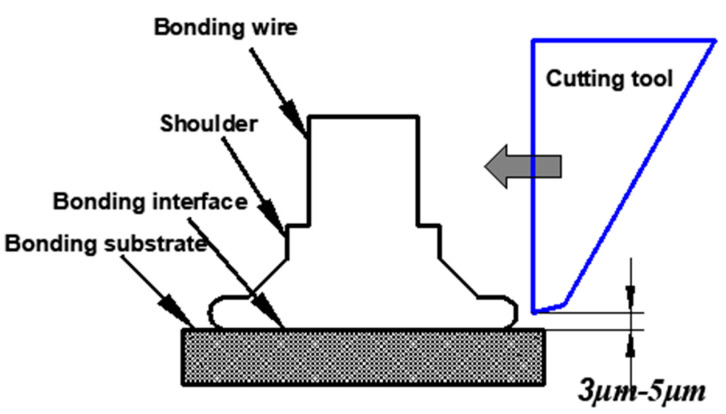
Schematic diagram of the ball bonded point shear test.

**Figure 3 micromachines-12-00024-f003:**
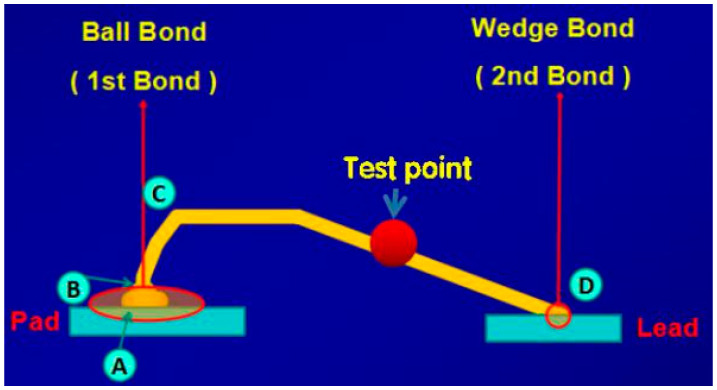
Schematic diagram of tensile test and breaking position.

**Figure 4 micromachines-12-00024-f004:**
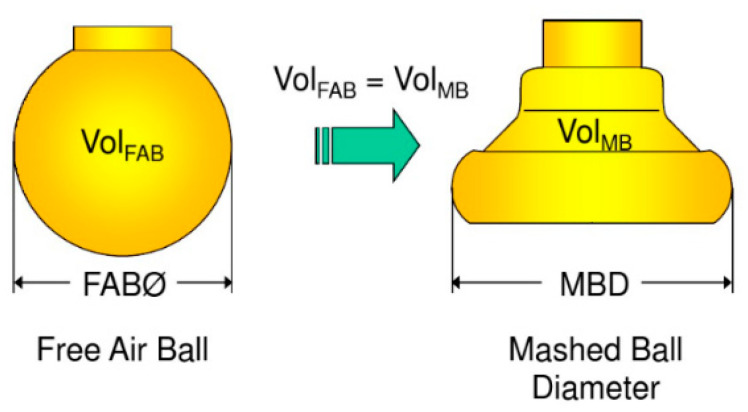
Comparison of the morphology of the ball bonded point before and after bonding.

**Figure 5 micromachines-12-00024-f005:**
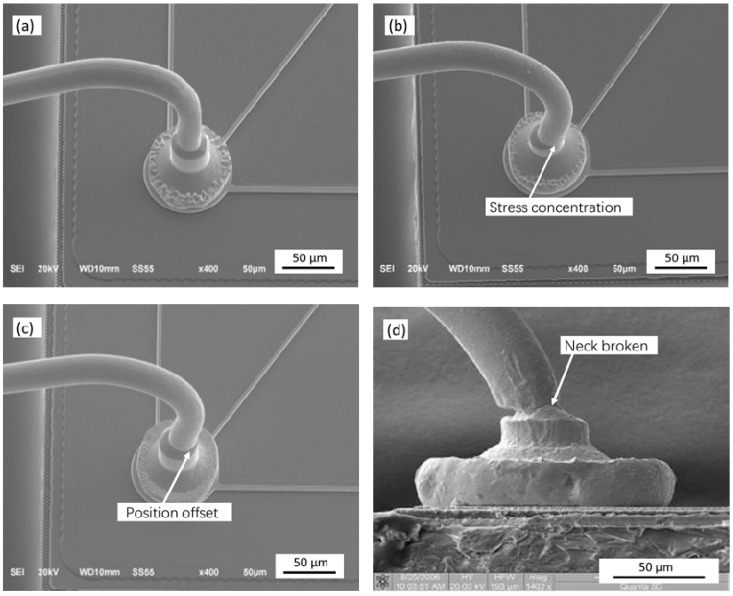
The morphology of the ball bonded points with the IHDs of: (**a**) 33 μm; (**b**) 26 μm; (**c**) 41 μm; (**d**) the neck fracture of the ball bonded point caused by the small IHD of the capillary.

**Figure 6 micromachines-12-00024-f006:**
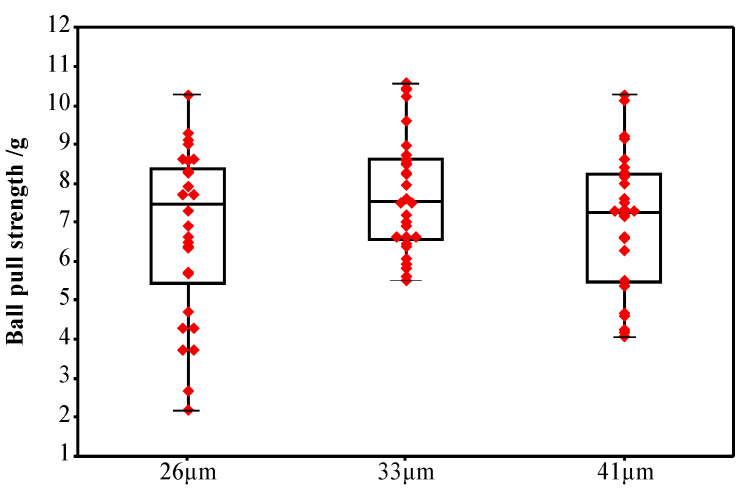
The pull strength of the ball bonded points with IHDs of 26, 33 and 41 μm respectively.

**Figure 7 micromachines-12-00024-f007:**
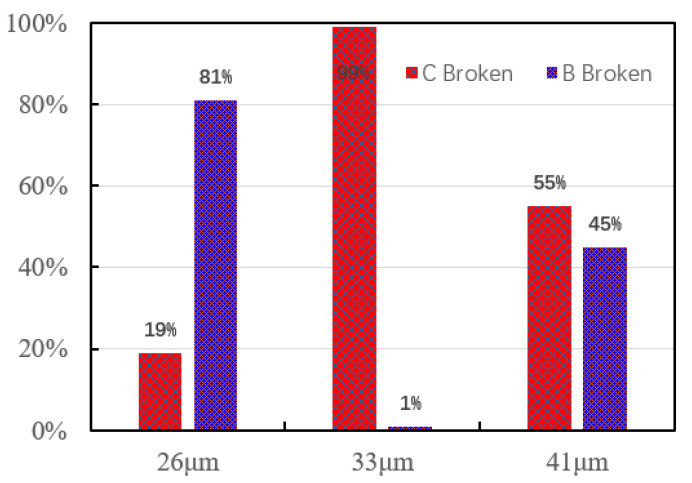
Statistical diagram of the breakpoint position of the ball bonded point with IHDs of 26, 33 and 41 μm respectively.

**Figure 8 micromachines-12-00024-f008:**
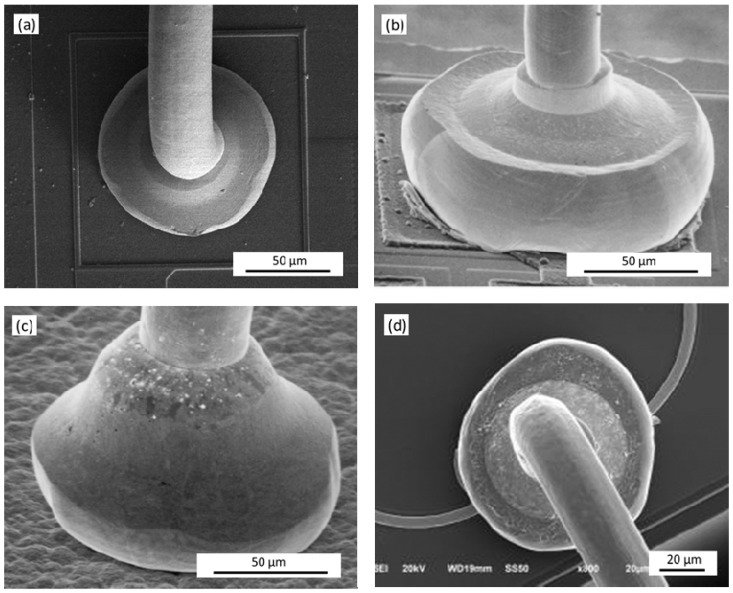
The shape of the ball bonded point with ICADs of: (**a**) 64 μm; (**b**) 51 μm; (**c**) 76 μm; (**d**) the MBD is too large and overflows the pad due to the small ICAD.

**Figure 9 micromachines-12-00024-f009:**
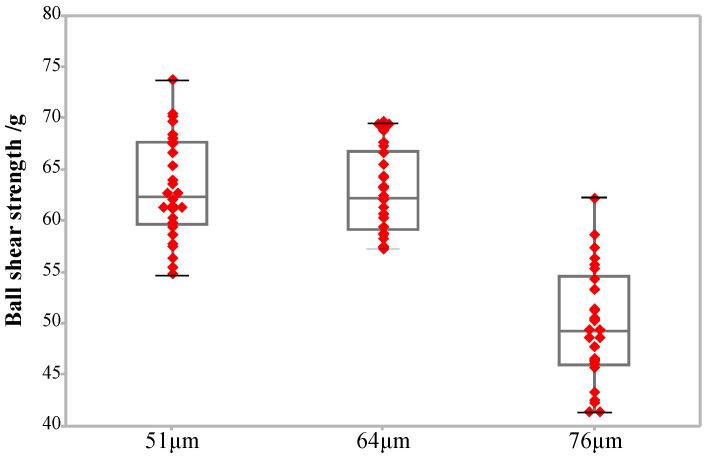
The shear strength of the ball bonded points when the ICAD is 51, 64 and 76 μm.

**Figure 10 micromachines-12-00024-f010:**
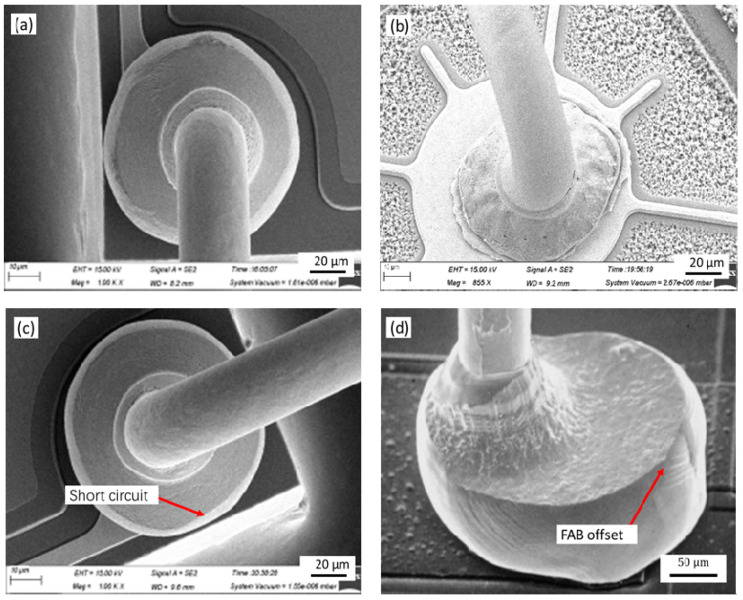
The morphology of the ball bonded points ICBAs of: (**a**) 100°; (**b**) 70°; (**c**) 120°; (**d**) the ICBA is so small that it forms an eccentric ball.

**Figure 11 micromachines-12-00024-f011:**
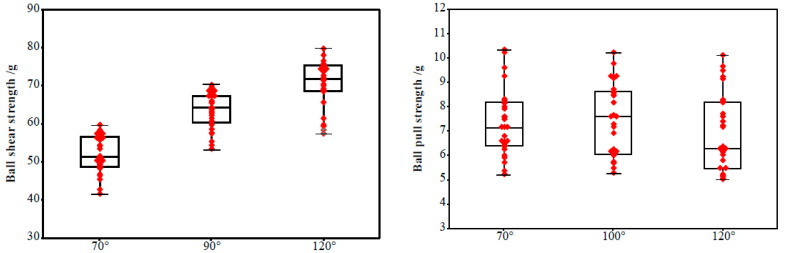
The shear strength and pull strength statistics of the ball bonded points with the different ICBA.

**Figure 12 micromachines-12-00024-f012:**
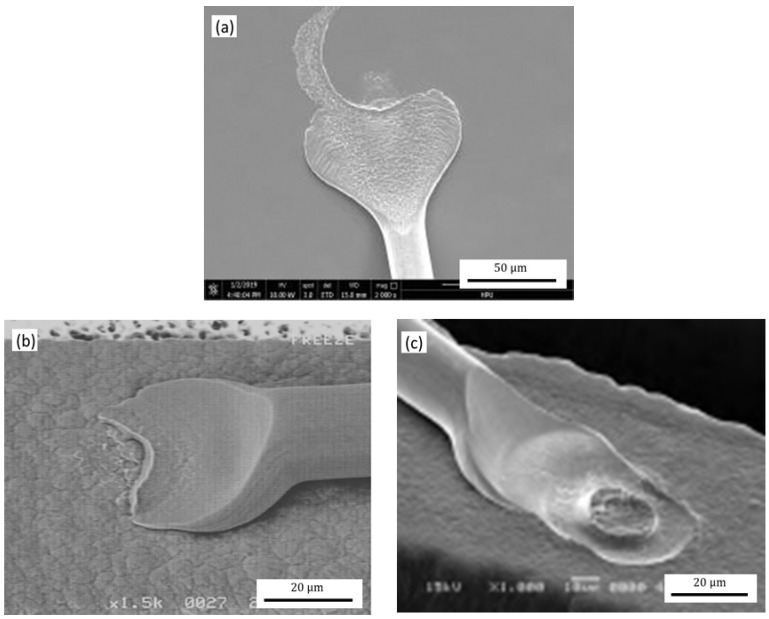
The morphology of the second bonded points with tip diameters (TDs) of: (**a**) 178 μm; (**b**) 127 μm; (**c**) 241 μm.

**Figure 13 micromachines-12-00024-f013:**
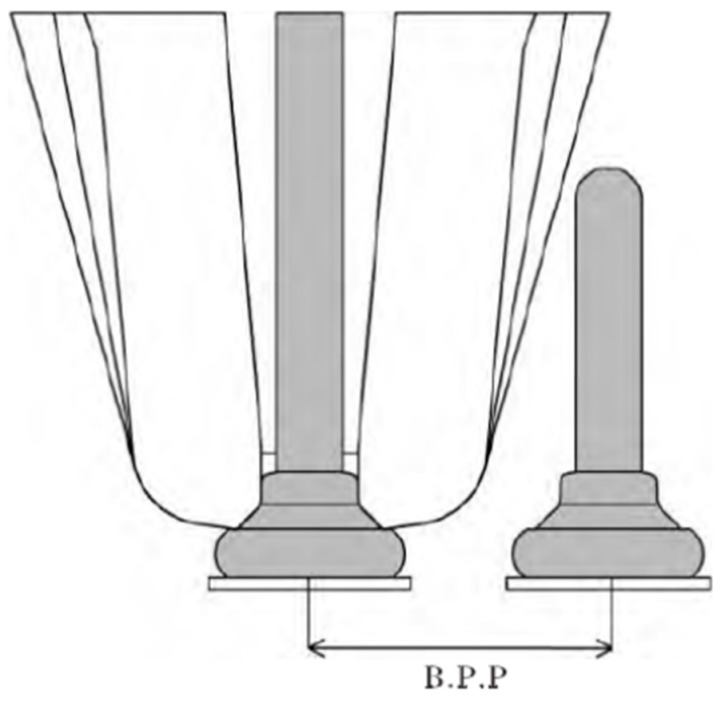
Schematic diagram of wire bonding pitch space of bonding pad.

**Figure 14 micromachines-12-00024-f014:**
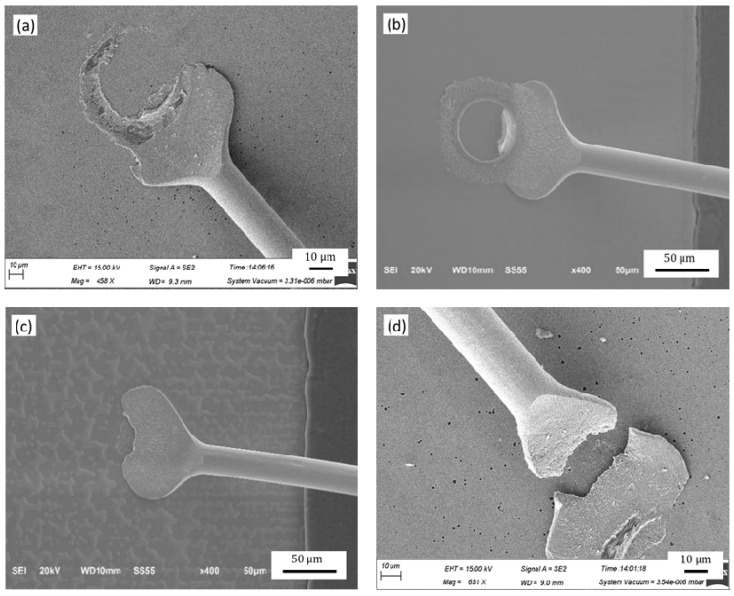
The topography of the second bonded points with WFAs of: (**a**) 8°; (**b**) 4°; (**c**) 11°; (**d**) second bonded point fracture due to the small WFA.

**Figure 15 micromachines-12-00024-f015:**
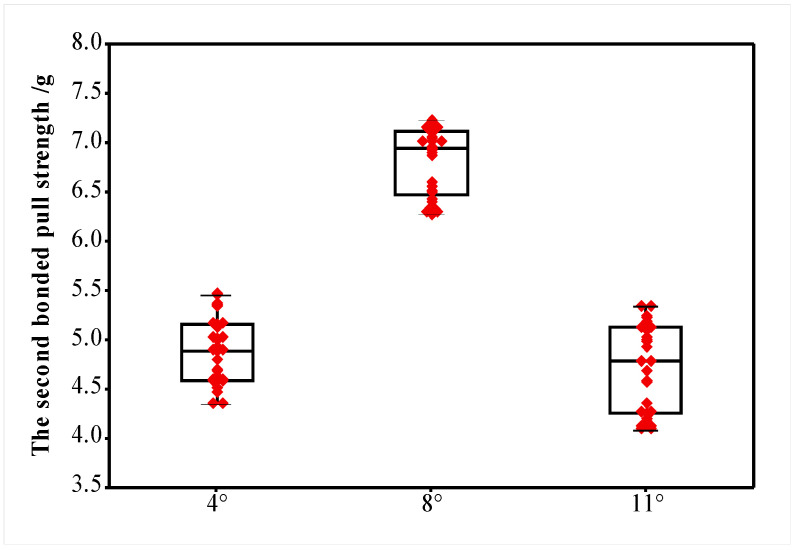
The pull strength of the second bonded point alloy wire when the WFA is 4°, 8° and 11°.

**Figure 16 micromachines-12-00024-f016:**
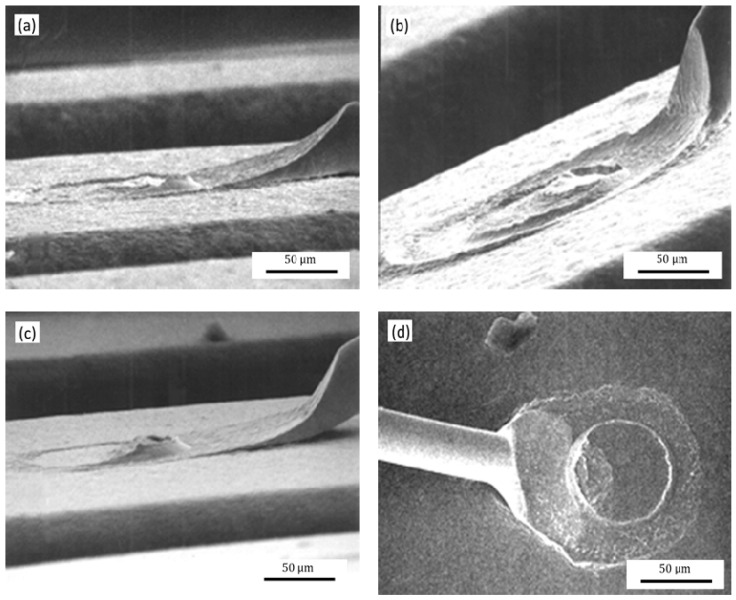
The morphology of the second bonded point when the OCR is: (**a**) 25.4 μm; (**b**) 20 μm; (**c**) 30 μm; (**d**) the second bonded point with small OCR.

**Figure 17 micromachines-12-00024-f017:**
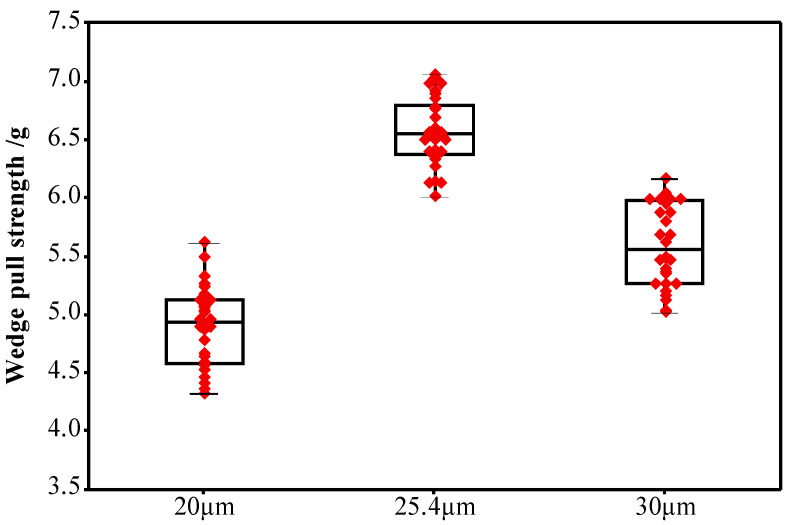
The tensile value of the second bonded points of alloy wires with OCRs of 20, 25.4 and 30 μm.

**Table 1 micromachines-12-00024-t001:** Alloy wire element content and mechanical properties table.

	Ag Content (%)	Au Content (%)	Strength (N)	Elongation (%)
Ag-5Au	95	5	0.095	14.1

**Table 2 micromachines-12-00024-t002:** Ceramic capillary geometric parameter table.

Items	IHD (μm)	ICAD (μm)	ICBA (°)	TD (μm)	WFA (°)	OCR (μm)
1	33	64	100	178	8	25.4
2	26	64	100	178	8	25.4
3	41	64	100	178	8	25.4
4	33	51	100	178	8	25.4
5	33	76	100	178	8	25.4
6	33	64	70	178	8	25.4
7	33	64	120	178	8	25.4
8	33	64	100	127	8	25.4
9	33	64	100	241	8	25.4
10	33	64	100	178	4	25.4
11	33	64	100	178	11	25.4
12	33	64	100	178	8	20.0
13	33	64	100	178	8	30.0

## Data Availability

The data presented in this study are available in article.

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
