# Peer review of "Effect of Ceramic Capillary Parameters on Bonded Morphology and Strength"

_micromachines, 2020, doi:10.3390/mi12010024_

Round 1

Reviewer 1 Report

1 Table 1, the unit of strengthen should be provided.

2 The distance between cutting tool and bonding substrate varies between 3-5 microns. There may be differences under this range which will lead to the variations of the tested solutions.

3 The language of this paper should be polished. Many typo errors can be found.

4 How to define the "wedge" used in this manuscript. It should be clearly defined. Similar comment to the wedge angle.

5 Too many figures, some of these figures can be assembled to be into one figure.

6 Any theoretical analysis could be provided by the authors to support the point 4 in the Conclusions?

7 The discussions on the failure mechanisms on bonding strengthen of this paper should be improved to avoid that it looks like a lengthy technical report.

Reviewer 2 Report

This is an informative paper.  Some good work has been carried out but the language has to be improved. Diagrams are generally well constructed.

Abstract

Line 11: "The influence of different geometric parameters of the ceramic capillary for the 12 morphology of φ25.4μmAg-5Au bonding alloy wire ball solder joints and wedge solder joints 13 are studied by using the automatic wire bonding machine, push-pull force tester, scanning 14 electron microscope etc." 

Consider breaking the sentence up to clearly identify the objectives. 

Line 17: Reduce 20 the inner cutting angle diameter to 51μm. The MBD is too large to overflow the pad, and the 21 inner cutting angle diameter is increased to 76μm. 

Consider rewording this sentence, the abstract should be informative and condensed. As such, the sentences should be better constructed. 

Line 47: Yao Li et al.[13]

There is a full stop after et al. Please consider removing it. This applies to the rest of the article. 

Table 1: Please reformat Table 1 properly. 

Line 147: consider changing the word "seriously reduced" to "heavily reduced"?

Line 148: "figure 9" should be "Figure 9" for consistency. 

Consider breaking up page 6 to a few more paragraphs and reformat it for emphasis. 

Figure 26: Please consider adding the scale bar for consistency.

Round 2

Reviewer 1 Report

All the comments have been responded accordingly. No further comments.